# Comparative Transcriptome Reveals Conserved Gene Expression in Reproductive Organs in Solanaceae

**DOI:** 10.3390/ijms26083568

**Published:** 2025-04-10

**Authors:** Lingkui Zhang, Yipeng Chong, Xiaolong Yang, Wenyuan Fan, Feng Cheng, Ying Li, Xilin Hou, Kang Zhang

**Affiliations:** 1State Key Laboratory of Crop Genetics and Germplasm Enhancement, College of Horticulture, Nanjing Agricultural University, Nanjing 210095, China; 82101219706@caas.cn (L.Z.); yingli@njau.edu.cn (Y.L.); 2State Key Laboratory of Vegetable Biobreeding, Key Laboratory of Biology and Genetic Improvement of Horticultural Crops of the Ministry of Agriculture and Rural Affairs, Sino-Dutch Joint Laboratory of Horticultural Genomics, Institute of Vegetables and Flowers, Chinese Academy of Agricultural Sciences, Beijing 100081, China; zhongyipeng@bua.edu.cn (Y.C.); yangxl@stu.qau.edu.cn (X.Y.); 82101232252@caas.cn (W.F.); chengfeng@caas.cn (F.C.)

**Keywords:** tomato, pepper, Solanaceae, YABBY, comparative transcriptome

## Abstract

The Solanaceae family, which includes key crops such as tomato, pepper, eggplant, wolfberry, and groundcherry, is distinguished by its diversity of fruit types. However, the conservation of gene expression regulatory networks across different species remains poorly understood. This study utilizes comparative transcriptomics to analyze 293 transcriptome samples from 22 Solanaceae species, focusing on the expression profiles of reproductive organ (flower and fruit)-specific genes. Our results reveal evolutionary conservation in the expression patterns of these genes, particularly within regulatory pathways essential for plant reproduction. A detailed comparative analysis of gene expression patterns between tomato and pepper reveals common regulatory networks governing fruit development. Furthermore, through co-expression network analysis, we identified functional partners of YABBY in flower/fruit development and found that YABBY genes coordinate fruit development through spatiotemporal dynamic expression, shaping its regulatory role. These findings provide valuable insights that can guide future research on fruit development genes in Solanaceae species.

## 1. Introduction

The Solanaceae family is crucial for global agriculture, containing many important fruit crops and diverse plant species [1,2,3]. This family includes major economically important crops such as tomatoes (*Solanum lycopersicum*), eggplants (*Solanum melongena*), and peppers (*Capsicum* spp.) [3,4]. Beyond these well-studied model species, the Solanaceae family has some lesser-known but valuable species, such as groundcherry (*Physalis pruinosa*) [5], wolfberry (*Lycium barbarum*) [6], and pepino (*Solanum muricatum*) [7], which primarily generate economic value through their fruit production. Although most Solanaceae fruits are classified as berries, they show striking differences in shape, size, and chemical composition both within species and across species [2,8,9,10,11,12]. Understanding flower and fruit development within this family is essential for agricultural productivity and enhancement of fruit quality.

Recent studies have significantly advanced our understanding of Solanaceae development, particularly in flowers and fruits [13,14,15,16]. Transcriptomic analyses at various developmental stages of flowers and fruits have been conducted in several Solanaceae species, including tomato [17,18,19,20], pepper [21,22,23,24], eggplant [25], and wolfberry [26,27], uncovering gene networks controlling key traits. These studies have identified important genes and molecular pathways associated with fruit maturation, size, and quality. For example, comparative analyses of time-series transcriptomes in pepper and eggplant have revealed genes and regulatory networks responsible for fruit shape variation by comparing distinct morphological variants [25,28]. Similarly, time-series transcriptome studies elucidated fruit ripening processes in tomato and pepper [29,30,31,32]. Integrated analyses combining gene expression and metabolite data have successfully linked specific genes to important compounds like wolfberry polysaccharides [33], pepper capsaicin [34,35], eggplant anthocyanin [36], and tomato carotenoids [19,37,38]. However, the evolutionary conservation of these regulatory networks across Solanaceae species requires further investigation.

Most transcriptomic studies have focused on single-species analyses, which limits our understanding of complex biological processes that are shared or uniquely evolved across species [39,40,41,42,43,44]. Comparative transcriptomics emerges as a powerful approach to address these limitations by interrogating gene expression across species [45,46,47,48,49]. Similarly to comparative genomics, which identifies conserved genes and regulatory regions, comparative transcriptomics provides deeper insights by revealing conserved gene expression patterns. This approach facilitates the identification of homologous genes with potentially similar functions and enables the identification of key genes involved in specific biological processes [46,50,51,52]. This method has proven effective in studying plant evolution, revealing conserved gene activity in roots and leaves of vascular plants [45,47,48,53], legume root nodules [54,55], and angiosperm pollen [46]. Furthermore, comparative transcriptomic approaches have been particularly effective in identifying co-expression networks and gene modules that exhibit remarkable consistency across different species [48,56,57,58,59]. These findings suggest that comparative transcriptomics can effectively distinguish between conserved and species-specific gene functions.

In this study, we conducted a comparative transcriptomic analysis across 22 Solanaceae species to explore the conservation and divergence of gene expression patterns, particularly in flower and fruit tissues. Through systematic comparison of organ-specific gene sets and developmental stage expression profiles, we identified evolutionarily conserved regulatory modules in floral and fruit organs. This comparative approach not only enhances our understanding of the molecular mechanisms governing fruit development but also lays the groundwork for future research into gene functions and evolutionary processes within the Solanaceae family.

## 2. Results

### 2.1. Constructing Multi-Organ Gene Expression Atlas with 293 Transcriptome Profiles Across 22 Solanaceae Species

We conducted a comprehensive comparative genomic analysis encompassing 22 Solanaceae species, including the economically significant crops such as tomato, potato, pepper, eggplant, tobacco, and wolfberry (Figure 1 and Appendix A). To construct a gene expression atlas, we collected and utilized a total of 293 mRNA samples from these species (Appendix A), which covered transcriptome profiles from various organs, including roots, stems, leaves, flowers, and fruits. The mRNA data were mapped to respective reference genomes and expression matrices with TPM (transcripts per million) values were generated (Section 4). To ensure data consistency, we averaged the TPM values across different biological replicates and developmental stages for each organ within each species. Since flowers and fruits are both reproductive organs, we combined the expression data from the flower and fruit tissues for each species. In this way, we obtained the expression profiles of root, stem, leaf, and flower/fruit across 22 Solanaceae species (Appendix A). Except for a few species undergoing additional polyploidization, the number of expressed genes (TPM > 2) in the Solanaceae species ranged from 19,331 to 24,355 (Appendix A). The proportion of expressed genes relative to the total gene count ranged from 0.41 to 0.70 across these species. Across the 22 species, flower/fruit exhibited the highest gene expression proportion (86.70%), which was followed by stems (81.54%), leaves (81.31%), and roots (80.63%) (Appendix A).

Organ-specific genes generally play a major role in organ establishment and functionality [46]. To identify organ-specific genes, we calculated the specificity measure (SPM) of each gene, which ranges from 0 (not expressed in an organ) to 1 (expressed only in the organ). We applied four progressive thresholds—top 2%, 5%, 10%, and 20% of SPM values—to comprehensively assess organ-specific gene expression patterns (Section 4). At the strictest threshold (top 2%), flower/fruit-specific genes constituted the highest proportion of all organ-specific genes (49.45%) across the 22 species, which was significantly higher than that observed in the other three organs (*p* < 2.72 × 10^−2^, Wilcoxon signed-rank test, one-tailed) (Figure 1 and Appendix A). As the threshold relaxes to top 5%, root-specific genes emerged as the most prevalent, closely followed by flower/fruit-specific genes. Expanding the threshold from top 2% to top 20% revealed a dynamic gene expression landscape: flower/fruit- and root-specific gene proportions decreased (flower/fruit: 49.45% to 26.72%; root: 37.27% to 27.53%), while stem- and leaf-specific gene proportions increased (stem: 5.20% to 22.00%; leaf: 8.09% to 24.17%). This pattern suggests that a higher number of genes are expressed with a high degree of specificity in flowers/fruits and roots.

### 2.2. Interspecies Comparison Highlights Conserved Flower/Fruit-Specific Gene Expression

To investigate the conservation of organ-specific gene expression across Solanaceae species, we conducted pairwise syntenic analysis on 22 species, identifying 3,641,574 syntenic orthologous gene pairs (Appendix A). The Jaccard similarity coefficient (JSC) was employed to measure the similarity between organ-specific gene sets across species. A JSC value of 1 indicates complete overlap of syntenic gene pairs in both sets, while a value of 0 signifies no overlap (Section 4). For each organ-specific gene set across the four types of organs, we compared these JSC values across different specific thresholds and different organ-specific gene sets (Figure 2a). In the different specific threshold comparisons, we observed that gene sets with high specificity (top 2%) exhibited substantially greater JSC values compared to low-specificity sets (top 20%) (Figure 2a). In organ-specific comparisons, flower/fruit-specific genes displayed the highest JSC values under high-specificity thresholds (2%) but showed reduced rankings below root and leaf genes at low-specificity sets (top 20%) (Figure 2a). This pattern indicates that a substantial number of genes with high levels of organ-specific expression are conserved across different Solanaceae species. The particularly high similarity observed among flower/fruit-specific genes at the top 2% threshold implies that these genes have played conserved roles in flower/fruit development throughout Solanaceae evolution.

We further performed functional enrichment analysis on 352 organ-specific gene sets (comprising four organs at four thresholds across 22 species) (Figure 2b). The most consistently enriched pathway was cell wall organization, appearing in 130 gene sets (Appendix A). Among TFs, the ERF (Ethylene Response Factor) family stood out as the most enriched, with 75 gene sets showing significant enrichment, particularly in root-specific expressed genes, which were represented in 57 gene sets (Appendix A). The shared pathways and TFs suggest that the associated genes may have evolved specialized functions in organ-specific contexts. In addition to this, distinct pathways and TFs were uniquely enriched for each organ. Flower/fruit-specific genes were predominantly enriched in pathways related to plant reproduction and cytoskeleton organization, with notable enrichment of TFs such as YABBY and MADS-box, both critical to floral development (Figure 2b). Specifically, MADS-box was enriched in 18 species, YABBY in 15 species, plant reproduction pathways in 13 species, and cytoskeleton-related genes in 9 species (Figure 2c). Leaf-specific genes were uniquely enriched in the photosynthesis pathway, consistent with their functional role. Stem-specific genes displayed unique enrichment only for the TALE transcription factor (Figure 2b). Root-specific genes did not display distinct enrichment for any unique pathways or transcription factors, though functional overlap was observed with other organs. For instance, redox homeostasis pathways were enriched in both root- and leaf-specific genes, while response-to-external-stimuli pathways, along with WRKY and ERF TFs, were enriched in both root- and stem-specific genes (Figure 2b). Overall, flower/fruit-specific expressed genes exhibited a broader and more diverse set of enriched pathways and transcription factors compared to those of root, stem, and leaf. This suggests a highly specialized regulatory network associated with flower/fruit development.

### 2.3. Temporally Synchronized Gene Expression Between Tomato and Pepper Throughout Fruit Development

Tomatoes and peppers are key Solanaceae crops in which the fruits serve as the primary economic organs. Our analysis revealed that both tomato and pepper exhibit high JSC values in their flower/fruit-specific expressed genes, with values of 0.93, 0.79, 0.59, and 0.37 at the top 2%, 5%, 10%, and 20% thresholds, respectively. These high values highlight the strong conservation of gene expression profiles in flower/fruit across tomato and pepper. To further explore this conservation, we examined transcriptome data from 10 flower/fruit developmental stages in both tomato and pepper. These stages included the flower buds, open flower, two periods of young fruit (YF), immature green and mature green fruit (MG), the breaking red stage (BR), three days after breaking red (BR), and mature fruit at seven and ten days after breaking red (MF) (Appendix A). Using the top 10% flower/fruit-specific genes identified earlier, we performed expression clustering across these developmental stages, incorporating data from roots, stems, and leaves for comparison (Section 4). This clustering analysis grouped the 5649 genes (2503 genes in tomato and 3076 genes in pepper) into six distinct clusters based on their expression patterns (Figure 3a and Appendix A). Each cluster corresponded to specific developmental stages: Cluster I was highly expressed in flower buds, Cluster II in open flowers, Cluster III in YF (two periods of young fruit), Cluster IV in MG (immature green and mature green fruit stages), Cluster V in the BR (breaking red stage and three days post-breaking red), and Cluster VI in MF (mature fruits at seven and ten days after breaking red) (Figure 3a). The numbers of genes grouped into six clusters varied, with a higher number of genes in Clusters I (1268, 22.37%) and II (1390, 23.09%), which corresponded to the flower stages (Figure 3a). We observed that the distribution of genes between tomato and pepper across these clusters was not uniform. In Cluster IV, which was associated with the immature green and mature green fruit stages, pepper had 2.83-fold more genes than tomato. Conversely, in Cluster V, linked to the breaking red and early post-breaking red stages, tomato had 2.09-fold more genes than pepper (Figure 3a).

JSC values between tomato and pepper clusters further underscored the conservation of gene expression profiles. Higher JSC values were observed between the same or closely related clusters in two species (Figure 3b). For instance, Cluster II in tomato has a higher JSC value with Clusters I (JSC = 0.20) and II (JSC = 0.67) in pepper. Similarly, higher JSC values were found between Cluster III in tomato and Clusters III (JSC = 0.21) and IV (JSC = 0.21) in pepper, as well as between Cluster V in tomato and Clusters V (JSC = 0.20) and VI (JSC = 0.16) in pepper. Our analysis showed that Cluster II exhibited the highest JSC (0.67) between tomato and pepper (Figure 3b). Functional enrichment analysis on the clusters revealed that the gene sets in these clusters, especially Cluster II, were consistently enriched in pathways related to plant reproduction and cytoskeleton organization, as well as in the TFs YABBY and MADS-box (Figure 3c). These TFs and pathways were previously identified as being specifically enriched in flower/fruit-specific genes, reinforcing the notion that these clusters are crucial for flower and fruit development. MADS-box transcription factors have been shown to be closely related to flower and fruit formation and development in different species of Solanaceae [60,61,62,63], which exhibit high evolutionary stability in Solanaceae as evidenced by our previous study, showing both low gene loss rates and highly conserved expression patterns [64]. These findings underscore the robust conservation of gene expression profiles between tomato and pepper, particularly in clusters corresponding to similar or successive developmental stages.

### 2.4. Co-Expression Network Analysis Reveals YABBY Gene Partners in Flower/Fruit Development and Polarity Regulation

A genome-wide co-expressed network was constructed in tomato. We adopted an algorithm that integrates both the Pearson’s Correlation Coefficient (PCC) and mutual rank (MR) to construct the co-expression network. A network containing 126,099 co-expression pairs among 24,426 genes was obtained for tomato. As shown in Figure 4a, visualization of the network revealed that genes specifically expressed in the same organ clustered together. Genes highly expressed in the flower/fruit (color-coded red in Figure 4a) were positioned at the center of the network, with many of them overlapping with other clusters. In this network, we found many interactions between MADS-box and *YABBY* genes. Based on previous classifications of YABBY and MADS-box gene families in tomato, we examined these interactions in detail [65,66]. Four YABBY genes exhibited co-expression with seven MADS-box genes, including the following: *SlCRCb* with *LeAP3*; *SlYABBY1a* with *SlAGL6* and *SlMBP2*; *SlYABBY2a* with *LeFUL2* and *TAGL2*; *SlYABBY5a* with *TM6*, *TM29*, and *SlMADS1* (Appendix A). These findings suggest that *YABBY* genes may collaboratively regulate flower and fruit development alongside MADS-box family members.

YABBY TFs play critical roles in both vegetative and reproductive development in plants and are pivotal to maintaining adaxial–abaxial polarity [67]. To more comprehensively investigate their functional partners, we applied a threshold (top 5% PCC, PCC > 0.57), identifying 6177 co-expression pairs involving 9 YABBY genes and 3361 interacting partners. Among these, *SlYABBY2a* exhibited the highest connectivity (1703 partners), while *SlCRCa* showed restricted interactions (418 partners) (Appendix A). The 3361 non-YABBY co-expressed genes were classified into 149 gene families based on their homologs in *Arabidopsis thaliana*. Of these families, 82 were significantly enriched (*p*-value < 0.05) in the YABBY co-expression network. These co-expressed gene families can be divided into six functional categories: TFs, cytoskeleton, cell cycle, signal transduction, metabolism, stress response and transport (Figure 4b). Our analysis showed that transcription factor families, including MADS-box, AP2, bHLH, and MYB were significantly enriched, highlighting potential regulatory interactions with YABBY genes (Figure 4b). The association between *AP2* and *SlYABBY2a* aligns with the ethylene-mediated fruit ripening process [68]. Additionally, BHLH and MYB TFs, known to influence floral and fruit development [69,70], are associated with *YABBY* genes. This result, consistent with co-expression network enrichment analyses in tomato and pepper, further confirms the collaborative regulatory roles of YABBY and these TFs in Solanaceae reproductive organ development. The network identified important functional partners functioning in cytoskeleton (*LOB* and *IQD* genes), cell cycle, signal transduction, metabolism, and stress response (Figure 4b), suggesting the involvement of YABBY genes in transcriptional regulation, organ development, signaling, and environmental adaptation.

### 2.5. Spatiotemporal Expression Dynamics of YABBY Genes Reveal Functional Specialization in Floral and Fruit Development

Although YABBY genes have been well-characterized in tomato flower and fruit development, their broader roles across the Solanaceae family remain largely unexplored [68,71,72,73]. Here, we identified 193 YABBY genes across 22 Solanaceae species (Appendix A). Using the 5 known YABBY genes from *Arabidopsis* and 9 YABBY genes present in tomato as a reference, we subdivided the 193 genes into nine distinct subfamilies: *CRCa* (20 genes), *CRCb* (21 genes), *INO* (22 genes), *YABBY1a* (24 genes), *YABBY1b* (21 genes), *YABBY2a* (28 genes), *YABBY2b* (15 genes), *YABBY5a* (22 genes), and *YABBY5b* (20 genes). These subfamilies primarily arose from whole-genome duplications, with paralogous relationships between *CRCa* and *CRCb*, *YABBY1a* and *YABBY1b*, *YABBY2a* and *YABBY2b*, and *YABBY5a* and *YABBY5b* (Figure 5a). Expression analysis revealed 167 out of 193 YABBY genes showed specificity in flower/fruit and leaf organs.

Expression patterns diverged between paralogous subfamilies: *YABBY2a* (82% flower/fruit-specific) versus *YABBY2b* (86.67% leaf-specific), and similarly for YABBY5a/5b. Overall, *CRCa*, *CRCb*, *YABBY1a*, *YABBY2a*, and *YABBY5a* were significantly enriched for flower-specific expression (*p* < 3.18 × 10^−2^), while *YABBY2b* and *YABBY5b* for leaf-specific expression (*p* < 1.75 × 10^−2^) (Figure 5a).

In tomato, pepper, eggplant, and wolfberry, nine YABBY genes were identified, corresponding to the nine subfamilies previously described. Our analysis identified that wolfberry shows a loss of the *CRCb* subfamilies and an additional *YABBY2b* gene copy due to tandem duplication (Figure 5b). All 36 YABBY genes in these four species were expressed in flowers, indicating a conserved role in flower development. The expression of *CRCa*, *CRCb*, and *INO* was highly conserved across species, with strong expression limited to flowers. The *YABBY1a, YABBY1b, YABBY2a, YABBY2b,* and *YABBY5a* were also expressed in leaf and fruit organs, indicating their roles in various organs. The *YABBY5b* genes displayed a distinct expression pattern, with expression confined to stems, leaves, and flowers but absent in fruit. To gain deeper insights into the fruit-specific functions of YABBY genes, we comprehensively analyzed the expression patterns of five fruit expressed genes—*SlYABBY1a*, *SlYABBY1b*, *SlYABBY2a*, *SlYABBY2b,* and *SlYABBY5a*—across various tomato fruit tissues (Appendix A). These genes were predominantly expressed in the outer and structural layers of the fruit, such as the septum, pericarp, inner epidermis, vascular tissue, parenchyma, collenchyma, and outer epidermis (Appendix A). *SlYABBY1a, SlYABBY1b,* and *SlYABBY5a* showed high expression levels during early fruit development (5 DAF and 10 DAF), while *SlYABBY2a* exhibited consistent expression throughout fruit development, especially in the septum and pericarp. *SlYABBY2b* was particularly prominent in the epidermis during all stages of fruit development (Appendix A). These expression patterns suggest that YABBY genes play an essential role in the development and structural integrity of fruit, particularly in maintaining the outer protective layers.

## 3. Discussion

The Solanaceae family encompasses a diverse array of crops, including tomato, pepper, eggplant, wolfberry, and groundcherry, where fruit plays a central role in economic value. Although tomato has emerged as a model system for studying fruit biology, it remains unclear to what extent its findings can be extrapolated to other Solanaceae species. Our comparative analysis of organ-specific gene expression across 22 Solanaceae species provides new insights into the evolutionary conservation of regulatory networks that govern flower and fruit development. Moreover, similar expression patterns of flower/fruit-specific genes were shared between the time-series transcriptome of tomato and pepper, which suggest that these genes are functionally conserved and likely inherited from a common ancestor.

Some studies highlighted the shared functions of key genes across Solanaceae species, which indicate the evolutionary conservation of flower and fruit regulatory networks. For example, *JOINTLESS* and *SFT* regulate flowering time in both tomato and pepper [74,75], while *BLIND* facilitates the transition to flowering in these species [76]. Genes such as *LOL1* (*LSD ONE LIKE1*) influence fruit color [77], *OFP20* (*Ovate Family Protein 20*) controls fruit shape [78], and *CNR/FW2.2* determines fruit size [79], as demonstrated in tomato, pepper, and groundcherry. These shared roles across species support the hypothesis of a conserved regulatory framework, but the cross-species conservation of floral and fruit development genes have not been comprehensively examined at the genome-wide level. In this study, the conservation of flower and fruit development genes across Solanaceae species is supported by high JSC between orthologous gene pairs, indicating inheritance from a common ancestor. Functional enrichment analyses further support this by revealing that flower/fruit-specific genes are consistently enriched in pathways related to reproduction and cytoskeletal organization, as well as key transcription factors such as YABBY and MADS-box. Although gene expression patterns can vary across developmental stages, the overall similarity in expression profiles during fruit development between tomato and pepper further highlights the conservation of these networks. These findings bolster the hypothesis that the regulatory networks governing flower and fruit development are conserved across Solanaceae species. The conservation of these genetic regulatory networks across Solanaceae species has significant implications for crop improvement strategies. Our findings provide a valuable framework for interspecies breeding approaches, where knowledge of gene function in one species can potentially be applied to improve traits in related crops. For instance, the identification of conserved regulatory genes like MADS-box genes and YABBY genes offers promising targets for molecular breeding programs aimed at optimizing fruit and flower development across multiple Solanaceae crops simultaneously.

We also investigated the conservation of YABBY genes across Solanaceae species through functional enrichment analysis of floral- and fruit-specific genes. In tomato, the YABBY gene family has been extensively studied in relation to flower and fruit development. Out of the nine subfamilies within the Solanaceae YABBY family, five have been functionally characterized in tomato (*CRCa*, *CRCb*, *INO*, *YABBY2a*, and *YABBY2b*), while four remain unresolved (*YABBY1a*, *YABBY1b*, *YABBY5a*, and *YABBY5b*). Specifically, *CRCa* has been shown to negatively regulate flower and fruit sizes in tomato, while *CRCb*, in conjunction with *CRCa*, controls flower development [71]. Additionally, *INO* plays a crucial role in ovule fate determination [80]. *YABBY2b* plays a role in controlling carpel number during flower and/or fruit development [81], while *YABBY2a* positively regulates fruit septum development and ripening [68]. However, the functions of these genes in other Solanaceae species remain largely unexplored. Comparative analysis of YABBY gene-specific expression revealed that five genes (*CRCa*, *CRCb*, *INO*, *YABBY2a*, and *YABBY2b*) previously characterized in tomato show generally conserved organ-specific expression patterns across 22 Solanaceae species. Additionally, these genes show similar temporal expression patterns in tomato, pepper, eggplant, and wolfberry. These findings suggest that researchers can confidently extend these results to the study of these five genes in non-tomato Solanaceae species. There are two paralogous gene pairs—*YABBY1a/1b* and *YABBY5a/5b*—that have not been fully functionally verified. Among these, *YABBY1b* exhibits similar expression patterns across species, while *YABBY1a* shows divergent expression patterns: it is expressed in the flowers and fruits of pepper and wolfberry, but only in the flowers of *Solanum* species like tomato and eggplant, indicating potential divergence in fruit-associated functions within the Solanaceae family. These results indicate that the YABBY gene family has preserved a conserved function throughout the evolutionary history of the Solanaceae family. Beyond the YABBY gene family, our analysis identified several other conserved gene modules that contribute to the evolutionary understanding of flower and fruit development across Solanaceae. The MADS-box gene family is another highly conserved transcriptional regulator group with critical roles in reproductive development. In our previous study, we found three MADS-box genes (*TM29*, *MADS-RIN*, and *SlMADS1*) with conserved expression patterns across Solanaceae species [64]. Functional analysis of *MADS-RIN* in pepper revealed similar roles in fruit ripening as observed in tomato, further supporting functional conservation [82]. Our current genome-wide analysis expands upon these findings, identifying additional MADS-box genes with conserved expression profiles.

Our genome-wide co-expression analysis revealed extensive interactions between YABBY genes and diverse functional partners. We identified interactions between YABBY and MADS-box genes, including *CRCb* with *LeAP3*; *YABBY1a* with *SlAGL6* and *SlMBP2*; *YABBY2a* with *LeFUL2* and *TAGL2*; and *YABBY5a* with *TM6*, *TM29*, and *SlMADS1.* These findings align with previous studies in Arabidopsis showing that during flowering transition, YABBY transcription factors interact with floral homeotic MADS-box proteins to establish flower primordia [83]. Our analysis also revealed associations between YABBY genes and genes encoding bHLH transcription factors, similar to cucumber where INO and bHLH protein (SPATULA) directly interact with *CRC* in fruit development [84]. The co-expression with MYB transcription factors supports findings in Arabidopsis where *FIL* activates *MYB28*, affecting secondary metabolite production [85]. Additionally, the co-expression of YABBY genes with KNOX (homeobox) genes highlights a developmental feedback loop where YABBY factors negatively regulate *KNOX* expression, influencing shoot apical meristem development and lateral organ formation [86]. Collectively, these conserved YABBY-centered regulatory networks provide valuable insights into flower and fruit development mechanisms in Solanaceae. Further experimental validation of these interactions will enhance our understanding of reproductive organ evolution and offer potential targets for crop improvement.

In summary, our analysis of organ-specific gene expression across 22 Solanaceae species demonstrates the evolutionary conservation of regulatory networks involved in flower and fruit development. The high conservation and organ-specificity of these genes highlight their critical roles in reproductive organ formation. While variations in expression patterns during fruit development were observed, the overall stability of these networks reinforces their importance in the evolution of reproductive structures within the Solanaceae family.

## 4. Materials and Methods

### 4.1. RNA-Seq Data Collection and Analysis

*Physalis angulata*, *Physalis floridana*, and *Lycianthes biflora* were cultivated in controlled laboratory conditions. Transcriptome sampling was initiated when plants reached ~30 cm in height. The other mRNA data from 22 Solanaceae species were collected from different organs (root, stems, leaf, flower, fruit) (Appendix A). The mRNA sequences for all species were mapped to their respective reference genomes (Appendix A) using HISAT2 v2.1.0 [87] (default parameters). The number of reads on each gene were counted by featureCounts v1.6.0 [88] (parameters “-p -F GTF”). The gene expression levels in all the samples were calculated using transcripts per kilobase of exon model per million mapped reads (TPM).

### 4.2. Identification of Organ-Specific Genes

Organ-specific genes were determined based on their expression profiles, using a method adapted from previous studies [46]. For each organ (e.g., root, stem, leaf, and flower/fruit), the TPM values from different biological replicates or developmental periods were averaged for subsequent analysis. For flower and fruit organs, transcriptome samples were averaged to represent both organs (flower/fruit) together. Genes with TPM value greater than 2 in at least one organ were selected for further analysis. Organ-specific genes were identified within each species individually using species-specific TPM values. To quantify organ specificity, we calculated the specificity measure (SPM) for each gene by dividing the average TPM for a given organ by the sum of the average TPM values across all organs. SPM values range from 0, indicating no expression in the sample, to 1, indicating the gene is entirely specific to that sample. To identify genes with sample-specific expression, for each of the 22 species, a threshold SPM value corresponding to the top 2%, top 5%, top 10%, and top 20% of SPM values was established [46,89,90,91].

### 4.3. Similarity of Organ-Specific Gene Groups Between Species

To assess the similarity of organ-specific transcriptomes across species, we calculated the JSC between sets of organ-specific genes. Syntenic gene pairs were identified between all potential pairs of 22 Solanaceous species. The SynOrths v1.5 tool [92] with default parameters (m = 20, n = 100, and r = 0.2) was used to perform syntenic gene identification. Organ-specific sets were identified by first determining the organ-specific genes in each species. Pairwise JSC values were then calculated for all gene sets. The JSC values range from 0 (completely distinct sets with no shared orthogroups) to 1 (identical sets of orthogroups). To determine if the organ-specific transcriptome of one species was significantly more similar to the corresponding organ in another species (e.g., comparing potato root to pepper root or tomato root), we tested whether the JSC values for the same organ across species were smaller (i.e., more similar) than those obtained from comparing the organ to different tissues (e.g., potato root versus pepper flower, or tomato leaf). The statistical significance of these comparisons was evaluated using the Wilcoxon rank-sum test.

### 4.4. Expression-Based Clustering of Genes in Tomato and Pepper

The expression matrices of all tomato flower/fruit-specific expressed genes and the expression matrices of pepper flower/fruit-specific expressed genes were combined into a single dataset. This dataset included expression profiles from roots, stems, leaves, flower buds, open flowers, and fruits across eight developmental stages in both species. Hierarchical clustering analysis was performed to identify gene expression patterns throughout flower and fruit development, using the R package ClusterGVis (https://github.com/junjunlab/ClusterGVis accessed on 1 October 2024). Six distinct clusters were identified with the getClusters function: Cluster I was highly expressed in flower buds, Cluster II in open flowers, Cluster III in YF (two periods of young fruit), Cluster IV in MG (immature green and mature green fruit stages), Cluster V in the BR (breaking red stage and three days post-breaking red), and Cluster VI in MF (mature fruits at seven and ten days after breaking red).

### 4.5. Functional Enrichment Analysis

The gene functions of the 22 Solanaceae species were annotated using the online tool Mercator4 v.2.0 (https://www.plabipd.de/ accessed on 1 October 2024). Transcription factors were identified using the online tool PlantTFDB v.5.0 (http://planttfdb.cbi.pku.edu.cn/ accessed on 1 October 2024) [93]. To gain insights into the biological processes associated with organ-specific gene expression, we performed functional enrichment analysis using Mercator4 terms and transcription factors. For each organ-specific gene set, enrichment was calculated using the hypergeometric test to determine over-represented function terms. The *p*-values for each enriched term were computed based on the hypergeometric distribution, reflecting the probability of observing a specific number of genes associated with a given Mercator4 term compared to the background gene set. To control for multiple testing, the Benjamini–Hochberg (BH) procedure was applied to adjust the *p*-values (padjust), ensuring a false discovery rate (FDR) below 0.05. Only Mercator4 terms and transcription factors with padjust values below this threshold were considered significantly enriched.

### 4.6. YABBY Gene Family Analysis

YABBY genes from 22 Solanaceae species were identified using PlantTFDB v.5.0. A phylogenetic tree was constructed by aligning the protein sequences of 193 YABBY genes from Solanaceae with those from *Arabidopsis thaliana*. The Neighbor-Joining method implemented in MEGA-X was employed to build the phylogenetic tree [94]. The organ of specific expression of the YABBY gene was identified using the top 10% list of specifically expressed genes. Expression of different tissues in tomato fruits downloaded from Tomato Expression Atlas (https://tea.solgenomics.net/, accessed on 1 October 2024).

### 4.7. Construction of the Co-Expression Networks

For the expression data of different tomato organs, we built a PCC matrix containing all possible gene pairs using R based on their normalized TPM values. Instead of selecting gene pairs with the highest PCC values, we applied an algorithm based on the mutual rank (MR) of PCC, a method previously used in constructing important co-expression databases such as ATTED-II [95]. This algorithm, along with its parameters, was thoroughly described and discussed in a prior study [96]. Specifically, we retained gene pairs with either MR top-3 or MR ≤ 30, combined with PCC ≥ 0.57, corresponding to the top 5% of the positive co-expression gene pairs. Gene family information for *Arabidopsis thaliana* was downloaded from TAIR (https://www.arabidopsis.org/download/list?dir=Genes, accessed on 1 October 2024). For tomato, gene families were identified by homologous comparison (best-hit, identity > 30%, e-value < 1 × 10^−5^) to *Arabidopsis* genes. Genes co-expressed with each YABBY gene were filtered and analyzed separately, and their associated gene families were identified. To assess gene family enrichment, we performed a hypergeometric test to determine the statistical significance of the enrichment for each set of co-expressed genes.

## Figures and Tables

**Figure 1 ijms-26-03568-f001:**
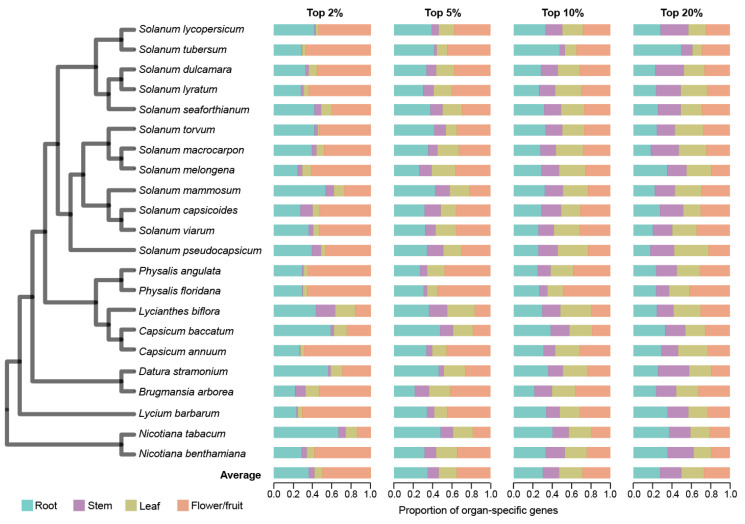
Characterization of organ-specific gene expression in each of the 22 Solanaceae species. The phylogenetic tree on the left illustrates the evolutionary relationships among 22 Solanaceae species. Organ-specific gene expression was assessed using four thresholds based on SPM value, with the top 2%, 5%, 10%, and 20% of genes ranked from highest to lowest SPM values selected for analysis. The bar chart displays the relative proportion of organ-specific genes across four organ types (root, stem, leaf, flower/fruit), showing the distribution of organ-specific genes as a percentage of the total number of organ-specific genes identified.

**Figure 2 ijms-26-03568-f002:**
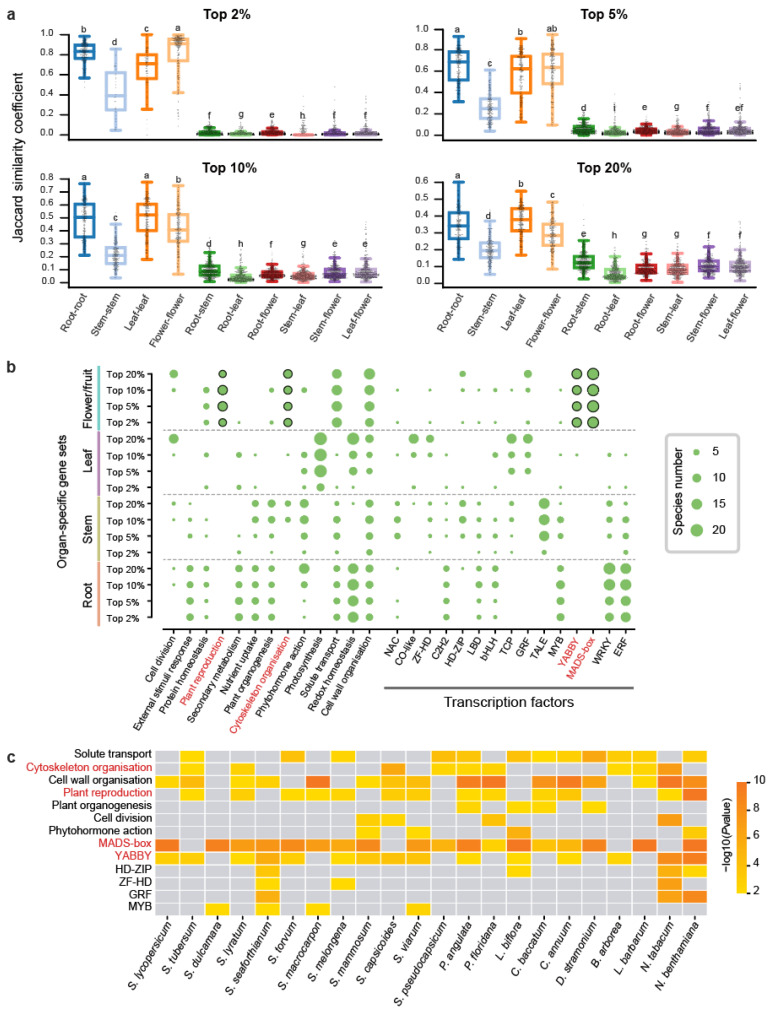
Comparison of organ-specific gene expression across 22 Solanaceae species. (**a**) Boxplot illustrating the Jaccard similarity coefficient (JSC) for organ-specific gene sets. The y-axis represents JSC values, while the x-axis compares the same organ (e.g., root–root) and one organ against others (e.g., root–stem). Higher JSC values indicate greater similarity between the transcriptomes. Statistical significance of differences (two-sided Mann-Whitney U test) is indicated by lettered annotations (*p* < 0.05). (**b**) Bubble plot depicting the number of species with significant functional enrichment (*p* < 0.05) for specific pathways and transcription factors. The vertical axis lists the 16 gene sets used for enrichment analysis, while the size of each bubble represents the number of species (out of 22) enriched for that particular pathway or transcription factor. (**c**) Heatmap visualization of enrichment results for the top 10% flower/fruit-specific genes across 22 Solanaceae species.

**Figure 3 ijms-26-03568-f003:**
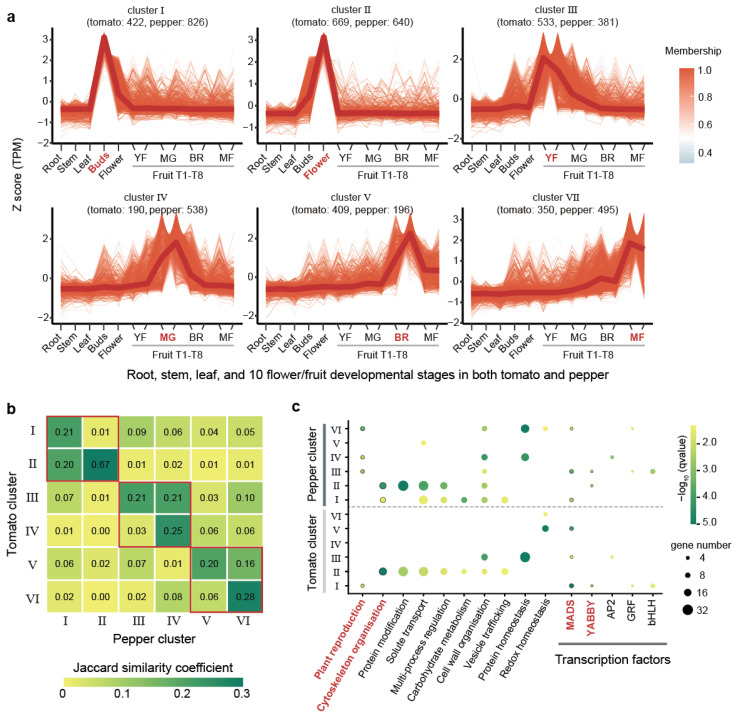
Expression clustering of floral/fruit time-series transcriptomes in tomato and pepper. (**a**) Clustering of flower/fruit-specific genes in tomato and pepper. The numbers in parentheses indicate the gene count for tomato and pepper within each cluster, respectively. Each line segment represents the expression pattern of a gene, with the color reflecting the gene’s expression trend based on the overall membership value in the cluster. Developmental stages in x-axis: YF (young fruit, two periods), MG (immature green and mature green fruit), BR (breaking red stage and three days post-breaking red), MF (mature fruit at seven and ten days post-breaking red). (**b**) Heatmap illustrating Jaccard similarity coefficient (JSC) values between clusters of tomato and pepper. The values compare the similarity between two of the six clusters from each species. The red boxes represent adjacent or identical periods of development. (**c**) Bubble plot showing significant functional enrichment (q-value < 0.05) for specific pathways and transcription factors. The bubble size corresponds to the number of genes involved, while bubble color represents the q-value of enrichment.

**Figure 4 ijms-26-03568-f004:**
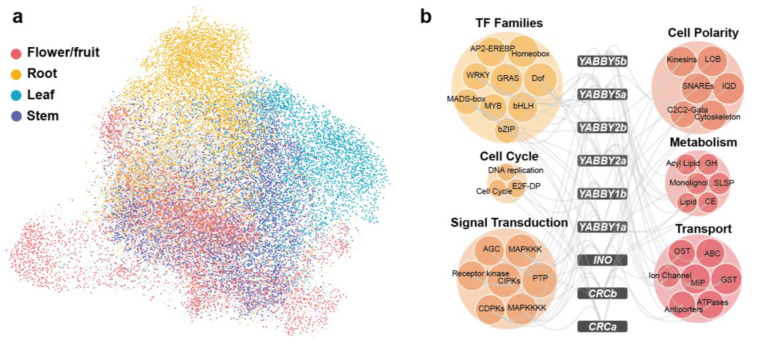
Gene clusters in the co-expression network built for tomato. (**a**) Overview of the co-expression network for tomato, with genes specifically expressed in different organs highlighted by distinct colors. The network was visualized using the “preferred layout” in Cytoscape v3.7.2. (**b**) Co-expressed gene families of the *YABBY* genes. The six colored bubbles represent six functionally classified gene families. Abbreviations for some gene families include DNA replication (core DNA replication machinery), cell cycle (core cell cycle genes), GH (glycoside hydrolase), SLSP (subtilisin-like serine proteases), monolignol (monolignol biosynthesis), CE (carbohydrate esterase), lipid (lipid metabolism), OST (organic solute cotransporters), and GST (glutathione S-transferase).

**Figure 5 ijms-26-03568-f005:**
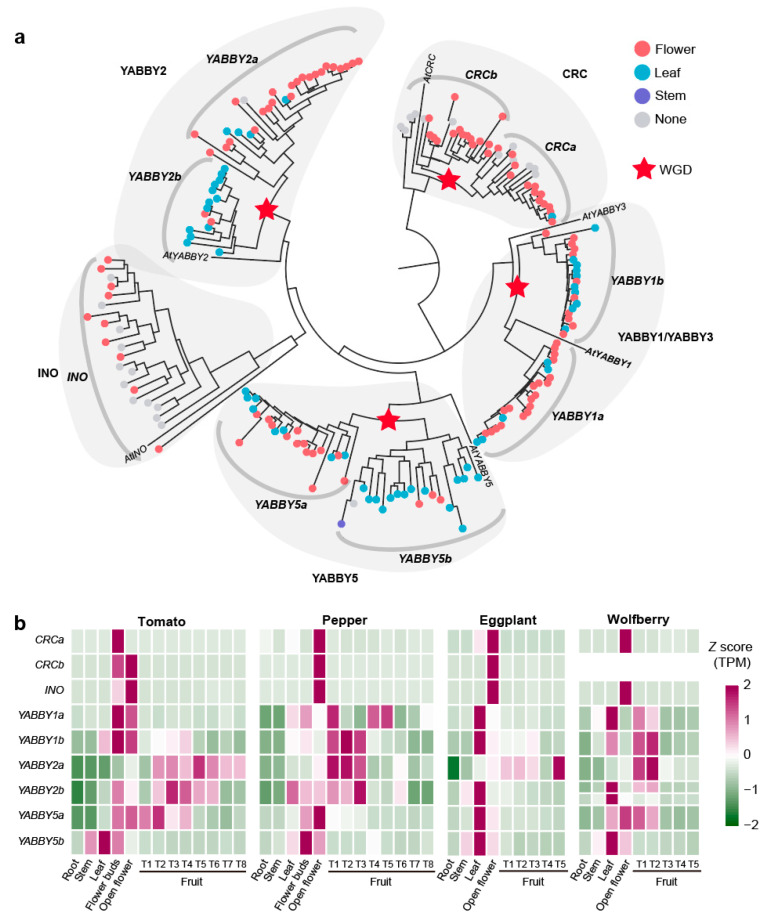
Evolutionary and expression characteristics of the YABBY gene family in Solanaceae. (**a**) Phylogenetic tree of YABBY genes across 22 Solanaceae species. The color of the circles on each branch indicates the gene’s organ-specific expression pattern: flower/fruit-specific, leaf-specific, stem-specific, or non-specific expression. (**b**) Heatmap displaying the expression levels of YABBY genes in tomato, pepper, eggplant, and wolfberry across different organs.

## Data Availability

The transcriptome data have been deposited in the National Genomics Data Center (NGDC, https://ngdc.cncb.ac.cn/, accessed on 1 October 2024) with the accession number BioProject PRJCA036112. The Sequence Read Archive (SRA) identifiers for publicly available transcriptome data used in this study are provided in Appendix A.

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
