# Peer review of "Comparative Transcriptome Reveals Conserved Gene Expression in Reproductive Organs in Solanaceae"

_ijms, 2025, doi:10.3390/ijms26083568_

Round 1

Reviewer 1 Report

Comments and Suggestions for Authors The manuscript "Comparative Transcriptome Reveals Conserved Gene Expression in Reproductive Organs in Solanaceae" focused on the expression profiles of reproductive organ-specific genes considering 22 Solanaceae species. The authors used a high number of samples. However, in the material and methods section, details about plant growth conditions and time of sampling for rna-seq are missing. In addition, details about the reference genomes used for mapping the reads are also missing. Please to provide these informations which, in my opinion, are essential in terms of full understanding and reproducibility of the research. The study is well-structured and provides valuable insights into the evolutionary conservation of gene expression patterns within the family. Particularly, the identification of YABBY gene partners involved in flower/fruit development is a significant finding. Focusing on the results section, Figures 2 and 3 could be enhanced with clearer labeling. Adding axis titles, improving color contrast, and including legends would improve readability. Consider providing an additional figure highlighting key pathways and transcription factors identified in flower/fruit-specific genes. Moreover, the identification of YABBY gene partners is interesting. However, the discussion does not address the potential biological implications of these interactions sufficiently. Consider to improve the discussions and make considerations also based on literature informations. Altogether, the study presents a valuable contribution to the understanding of gene expression conservation in Solanaceae. With appropriate revisions, the manuscript will be suitable for publication in the ijms

Author Response

Comments 1: The manuscript "Comparative Transcriptome Reveals Conserved Gene Expression in Reproductive Organs in Solanaceae" focused on the expression profiles of reproductive organ-specific genes considering 22 Solanaceae species. The authors used a high number of samples. The study is well-structured and provides valuable insights into the evolutionary conservation of gene expression patterns within the family. Particularly, the identification of YABBY gene partners involved in flower/fruit development is a significant finding.

Response 1: We would like to express our sincere gratitude for your time and effort in reviewing our manuscript. We appreciate your constructive feedback and valuable suggestions, which have significantly improved the quality of our manuscript. Below, we have addressed each of your comments and concerns point by point.

Comments 2: However, in the material and methods section, details about plant growth conditions and time of sampling for rna-seq are missing. In addition, details about the reference genomes used for mapping the reads are also missing. Please to provide these informations which, in my opinion, are essential in terms of full understanding and reproducibility of the research.

Response 2: We thank the reviewer for pointing out these important omissions. In the revised manuscript, we have added detailed descriptions of the plant growth conditions and sampling times used for RNA-seq in the Materials and Methods section. Additionally, we have provided a comprehensive list of reference genomes used for read mapping in Supplementary Table S1.

Comments 3: Focusing on the results section, Figures 2 and 3 could be enhanced with clearer labeling. Adding axis titles, improving color contrast, and including legends would improve readability.

Response 3: We appreciate the reviewer's suggestion regarding the improvement of Figures 2 and 3. We have revised these figures accordingly.

Comments 4: Consider providing an additional figure highlighting key pathways and transcription factors identified in flower/fruit-specific genes.

Response 4: Following the reviewer's valuable suggestion, we have added a new figure panel (now Figure 4c) that highlights key pathways and transcription factors identified in flower/fruit-specific genes across 22 species.

Comments 5: Moreover, the identification of YABBY gene partners is interesting. However, the discussion does not address the potential biological implications of these interactions sufficiently. Consider to improve the discussions and make considerations also based on literature informations. Altogether, the study presents a valuable contribution to the understanding of gene expression conservation in Solanaceae. With appropriate revisions, the manuscript will be suitable for publication in the ijms

Response 5: We thank the reviewer for encouraging us to expand this important aspect of our work. We have substantially revised the Discussion section to elaborate on the biological implications of the identified YABBY gene partners as suggested.

Reviewer 2 Report

Comments and Suggestions for Authors

The study titled "Comparative Transcriptome Reveals Conserved Gene Expression in Reproductive Organs in Solanaceae" presents an in-depth comparative transcriptomic analysis of 22 species within the Solanaceae family, focusing on gene expression in reproductive organs such as flowers and fruits. Using 293 transcriptome samples, the authors explore organ-specific gene expression, highlight conserved regulatory modules, and provide a detailed analysis of the YABBY gene family and its co-expression networks. The study is well-structured, methodologically sound, and provides relevant insights into the evolutionary conservation of gene expression patterns in economically important Solanaceae crops. However, I have comment to improve the document.

There are minor grammatical issues that should be addressed to improve the fluency of the manuscript. Examples include:
Line 55: "Most transcriptomic studies has..." → should be "have".
Line 68: "Comparative transcriptome can..." → should be "Comparative transcriptomics can".
Line 233: "Network was contructed..." → should be "constructed".

The manuscript occasionally overuses expressions like "Interestingly", "Remarkably", and "Notably", which could be limited to maintain a more neutral scientific tone.

The Results section is lengthy and dense, especially sections 2.3–2.5. Consider summarizing portions of the analysis or moving some descriptive parts to supplementary materials.

Certain statements and findings are repeated unnecessarily, particularly concerning the conservation of YABBY gene expression patterns. These should be streamlined for conciseness.
In M&M, while TPM normalization is appropriate, the manuscript lacks detail on how inter-species normalization was handled, particularly considering differences in genome size and gene content. Clarifying this would strengthen the validity of the comparative approach.
The rationale for selecting the top 2%, 5%, 10%, and 20% thresholds for SPM could be better supported with references or preliminary analysis.

The Discussion focuses heavily on the YABBY gene family, which is indeed a highlight of the study. However, expanding the analysis to include other conserved genes or modules would enrich the evolutionary perspective.
Consider discussing more applied implications, such as how these findings might contribute to breeding programs or gene editing strategies across Solanaceae crops.

Comments on the Quality of English Language

There are minor grammatical issues that should be addressed to improve the fluency of the manuscript. Examples include:
Line 55: "Most transcriptomic studies has..." → should be "have".
Line 68: "Comparative transcriptome can..." → should be "Comparative transcriptomics can".
Line 233: "Network was contructed..." → should be "constructed".

The manuscript occasionally overuses expressions like "Interestingly", "Remarkably", and "Notably", which could be limited to maintain a more neutral scientific tone.

Author Response

Comments 1: The study titled "Comparative Transcriptome Reveals Conserved Gene Expression in Reproductive Organs in Solanaceae" presents an in-depth comparative transcriptomic analysis of 22 species within the Solanaceae family, focusing on gene expression in reproductive organs such as flowers and fruits. Using 293 transcriptome samples, the authors explore organ-specific gene expression, highlight conserved regulatory modules, and provide a detailed analysis of the YABBY gene family and its co-expression networks. The study is well-structured, methodologically sound, and provides relevant insights into the evolutionary conservation of gene expression patterns in economically important Solanaceae crops. However, I have comment to improve the document.

Response 1: We sincerely appreciate your thorough review of our manuscript. Thank you for acknowledging the strengths of our study and providing constructive feedback to improve its quality. We have carefully addressed each of your comments as detailed below.

Comments 2: There are minor grammatical issues that should be addressed to improve the fluency of the manuscript. Examples include:
Line 55: "Most transcriptomic studies has..." → should be "have".
Line 68: "Comparative transcriptome can..." → should be "Comparative transcriptomics can".
Line 233: "Network was contructed..." → should be "constructed".
Response 2: We appreciate the reviewer for identifying these grammatical issues. The manuscript has been carefully proofread, and all grammar and typographical errors—including those cited (e.g., Lines 55, 68, 233)—have been corrected for accuracy and consistency.

Comments 3: The manuscript occasionally overuses expressions like "Interestingly""Remarkably", and "Notably", which could be limited to maintain a more neutral scientific tone.
Response 3: Thank you. We have reviewed the manuscript and removed or revised excessive uses of evaluative adverbs.

Comments 4: The Results section is lengthy and dense, especially sections 2.3–2.5. Consider summarizing portions of the analysis or moving some descriptive parts to supplementary materials. Certain statements and findings are repeated unnecessarily, particularly concerning the conservation of YABBY gene expression patterns. These should be streamlined for conciseness.
Response 4: We agree with the reviewer’s assessment and have taken steps to improve conciseness. Redundant descriptions—particularly those concerning YABBY gene conservation—have been removed.

Comments 5: In M&M, while TPM normalization is appropriate, the manuscript lacks detail on how inter-species normalization was handled, particularly considering differences in genome size and gene content. Clarifying this would strengthen the validity of the comparative approach.

Response 5: Thank you for this important methodological observation. To clarify, we did not perform direct cross-species normalization of gene expression. Instead, TPM values were independently calculated within each species. Organ-specific genes were identified within species, and the interspecies comparative analyses were conducted based on conserved ortholog groups. This strategy follows the approach used in a previous study published in Nature Plants [1], where TPM values of orthologous genes were analyzed without cross-species normalization. We have now clarified this procedure in the Materials and Methods section.

Relevant studies:

[1] Comparative transcriptomic analysis reveals conserved programmes underpinning organogenesis and reproduction in land plants. Nature Plants 2021.

Comments 6: The rationale for selecting the top 2%, 5%, 10%, and 20% thresholds for SPM could be better supported with references or preliminary analysis.
Response 6: We appreciate the reviewer’s insightful comment regarding the rationale for selecting multiple SPM thresholds (2%, 5%, 10%, and 20%). To address this concern, we have expanded the explanation in the Methods section to better justify our approach. Previous studies have adopted a range of SPM thresholds to identify tissue- or organ-specific genes, including top 5% [1,2], top 10% [3], and top 15% [4], demonstrating the lack of a universally fixed cutoff. Therefore, the use of multiple thresholds in our analysis was purposefully implemented to capture a gradient of expression specificity. Since organ-specific expression is inherently continuous rather than binary, examining multiple percentile cutoffs allows for a more nuanced and comprehensive characterization of specificity across different expression levels.  

Relevant studies:

[1] Comparative transcriptomic analysis reveals conserved programmes underpinning organogenesis and reproduction in land plants. Nature Plants 2021.

[2] Genome-Wide Discovery of Tissue-Specific Genes in Maize. Plant Molecular Biology Reporter 2016.

[3] Circadian-driven tissue specificity is constrained under caloric restricted feeding conditions. Communications biology 2024.

[4] Exploiting plant transcriptomic databases: Resources, tools, and approaches. Plant Communications 2022.

Comments 7: The Discussion focuses heavily on the YABBY gene family, which is indeed a highlight of the study. However, expanding the analysis to include other conserved genes or modules would enrich the evolutionary perspective.

Response 7: We agree with the reviewer that expanding our discussion beyond the YABBY gene family would enrich the evolutionary perspective of our study. We have substantially revised the Discussion section to include: Discussion of MADS-box transcription factors and their conserved co-expression modules in flower development.

Comments 8: Consider discussing more applied implications, such as how these findings might contribute to breeding programs or gene editing strategies across Solanaceae crops.

Response 8: We thank the reviewer for suggesting the inclusion of more applied implications of our findings. We have added a new subsection to the Discussion that addresses: Potential applications for crop improvement through targeted breeding strategies focusing on conserved regulators of flower and fruit development.

Reviewer 3 Report

Comments and Suggestions for Authors

This study has been reviewed. However, there are correctable errors in which the authors should write the respective gene correctly so that the reader is not confused.

Author Response

Comment: This study has been reviewed. However, there are correctable errors in which the authors should write the respective gene correctly so that the reader is not confused.

Response: We thank the reviewer for highlighting the errors in gene nomenclature that could potentially confuse readers. We have thoroughly reviewed the entire manuscript and have implemented corrections to ensure consistency with standard gene naming conventions.